# Targeting Lysophosphatidic Acid in Cancer: The Issues in Moving from Bench to Bedside

**DOI:** 10.3390/cancers11101523

**Published:** 2019-10-10

**Authors:** Yan Xu

**Affiliations:** Department of Obstetrics and Gynecology, Indiana University School of Medicine, 950 W. Walnut Street R2-E380, Indianapolis, IN 46202, USA; xu2@iu.edu; Tel.: +1-317-274-3972

**Keywords:** Autotaxin (ATX), ovarian cancer (OC), cancer stem cell (CSC), electrospray ionization tandem mass spectrometry (ESI-MS/MS), G-protein coupled receptor (GPCR), lipid phosphate phosphatase enzymes (LPPs), lysophosphatidic acid (LPA), phospholipase A_2_ enzymes (PLA_2_s), nuclear receptor peroxisome proliferator-activated receptor (PPAR), sphingosine-1 phosphate (S1P)

## Abstract

Since the clear demonstration of lysophosphatidic acid (LPA)’s pathological roles in cancer in the mid-1990s, more than 1000 papers relating LPA to various types of cancer were published. Through these studies, LPA was established as a target for cancer. Although LPA-related inhibitors entered clinical trials for fibrosis, the concept of targeting LPA is yet to be moved to clinical cancer treatment. The major challenges that we are facing in moving LPA application from bench to bedside include the intrinsic and complicated metabolic, functional, and signaling properties of LPA, as well as technical issues, which are discussed in this review. Potential strategies and perspectives to improve the translational progress are suggested. Despite these challenges, we are optimistic that LPA blockage, particularly in combination with other agents, is on the horizon to be incorporated into clinical applications.

## 1. Introduction

Lysophosphatidic acids (LPAs) are simple lipids, but they are involved in virtually every aspect of tumor development, covering all 10 cancer hallmark activities [1,2,3,4,5,6,7]. They include, but are not limited to, stimulation of the proliferative signaling [8,9,10], evading growth suppressors and resisting cell death by regulating the apoptotic and other cell death/survival pathways [11,12], enabling replicative immortality by regulating telomerase [13], inducing angiogenesis and lymphangiogenesis via upregulation of proangiogenic factors, such as vascular endothelial growth factor A (VEGFA), vascular endothelial growth factor C (VAGFC), interleukin (IL)-1β [14,15,16,17,18] and IL-8 [19,20,21], and activating invasion and metastasis [22,23,24,25,26]. In addition, LPA affects genome instability (the autotaxin (ATX)-LPA axis is involved in reactive oxygen species (ROS)-induced genomic instability [27] and γ-irradiation-induced DNA damage repair [28], inflammation by regulating inflammatory factors, such as cyclooxygenase-2 (COX2), IL6, and Tumor necrosis factor-alpha (TNFα) [29,30,31]) and energy metabolism (ATX–LPA signaling contributes to obesity-induced insulin resistance [32]; LPA triggers glycolytic shift and induces metabolic reprogramming in ovarian cancer via Rac-mediated activation of nicotinamide adenine dinucleotide phosphate oxidase (NADPH oxidase) and generation of reactive oxygen species (ROS), resulting in activation of hypoxia inducible factor 1-alpha (HIF1α) [33] and the immune system [1,2,3,4,5,6,7,34]). In particular, expression of ATX or each of the endothelial-derived G-protein–coupled receptor (EDG)-family LPA receptor (LPAR) (LPAR_1–3_) in the mammary epithelium of transgenic mice was shown to be sufficient to induce breast cancer [26]. The current review does not cover these functions or signaling pathways in detail, but rather focuses on the issues pertinent to the translation of LPA targeting to clinic applications. The different aspects of LPA’s functions in cancer were extensively reviewed over several decades [3,4,5,7,21,35,36,37,38,39,40,41,42,43,44,45,46,47,48,49].

The major challenges in moving LPA targeting to clinical practice include the complex metabolic network of LPA, the extremely broad and multifaceted pathological activities elicited by LPA, which are overlapping with its physiological activities, the complicated and potentially opposing cellular activities mediated by different LPA receptors in an individual cancer and individual patient-dependent manner, and the maze of intertwined G-protein coupling and downstream signaling pathway elicited by LPA through its own receptors, as well as many other types of cell receptors and signaling pathways. Moreover, technical issues for LPA detection and/or blockage, as well as study design issues are also major obstacles to overcome. This review focuses on these issues with perspectives to improve the LPA translational progress.

## 2. LPA: From Bench to Bedside

### 2.1. A Brief History and Milestones of LPA Research

#### 2.1.1. Before the Identification of LPA Receptors

LPA was first described in the early 1960s [50,51]. It was later studied in almost all type of cells in organisms ranging from bacteria to plants to animals. In the 1960s, only eight papers were published related to LPA. This number increased to ~40, ~100, and ~300 in the 1970s, 1980s, and 1990s, respectively. At the turn of the century, the number of LPA-related studies increased exponentially to more than 4000 papers related to LPA published since the year 2000 (Figure 1). Among these LPA papers, ~1300 papers are related to cancer/tumor, covering almost all types of solid and blood cancers/tumors (Figure 1). This list includes, but is not limited to, cancers of ovarian, lung, gastric, colorectal, breast, prostate, bladder, endometrial, renal, oral, pancreatic, cervical, and brain (including glioblastoma) origin, as well as leukemias, non-Hodgkin’s lymphomas, fibrosarcoma, osteosarcoma, and melanoma [3,4,21,48,52]. Among these publications, ~20% of the papers are related to ovarian cancer (OC) alone.

LPA was first isolated from brain extracts in 1961 [50]. Most studies on LPA at the early stage (from 1960s to 1970s) focused on the biochemical analyses of LPA, including enzymes involved in LPA metabolism and catabolism, as well as the tissue, cellular, and sub-cellular locations of LPA. Several of these studies are structure–activity studies, as LPA is a group of more than 20 molecules, varying in their fatty-acid chain location (sn-1 vs. sn-2 on the glycerol backbone), the numbers of carbons in the fatty-acid chain, the number and location of the double bonds in the fatty-acid chain, and also the chemical linkage between the fatty acid and the glycerol (ether linkage vs. ester linkage) [96,97,98].

The 1980s were the beginning decade for extensive functional and signaling studies of LPA. LPA-induced platelet aggregation and alterations in arachidonate metabolism were the earliest LPA functions revealed, which were further studied over the following decades [31,96,99,100,101,102,103,104]. The effects of LPA on ion channels were noticed as early as the 1980s [98] and regained more interest in recent years [98,105,106,107,108,109,110]. The intravenous injection of LPA induces hypertension in animals [97,111,112,113]. The mitogen (cell proliferation) activity of LPA was discovered in the late 1980s and early 1990s, before the molecular cloning of LPA receptors [35,53,54,55,56]. In addition, the potent effects of LPA on cell skeleton-related activities were reported in various cell types and/or organisms even before its receptors were identified/cloned in 1996. LPA induces contraction of rat isolated colon [114], reverts the β-adrenergic agonist-induced morphological response in C6 rat glioma cells [115], induces neuronal shape changes [57], and is a chemoattractant for *Dictyostelium discoideum* amoebae and human neutrophils [58,116]. Moreover, LPA inhibits gap-junctional communication and stimulates phosphorylation of connexin-43 in while blood cells. Focal adhesion kinase (FAK), paxillin, and p130 are important LPA-targeting genes/proteins [117,118,119].

In terms of signaling properties, calcium and cyclic adenosine monophosphate (cAMP) are the earliest revealed downstream signaling molecules for LPA [120]. Its regulation of protein phosphorylation was also discovered [117,118,119,121]. One of the milestone papers for LPA signaling published by Moolenaar’s group in the late 1980s showed that LPA initiates at least three separate signaling cascades: activation of a pertussis toxin-insensitive G-protein mediating phosphoinositide hydrolysis with subsequent Ca^2+^ mobilization and stimulation of protein kinase C; release of arachidonic acid in a guanosine triphosphate (GTP)-dependent manner, but independent of prior phosphoinositide hydrolysis; and activation of a pertussis toxin-sensitive G_i_-protein mediating inhibition of adenylate cyclase [53]. Later, the same group of investigators identified Ras activation as an important downstream signaling pathway for LPA in fibroblasts [59,119]. Another important finding is that the cell skeleton effects of LPA are linked to the small GTP-binding protein Rho [122].

The implications of the potential roles of LPA in cancer stem from the findings for LPA’s mitogen- and growth factor-like activity in the late 1980s and early 1990s [35,53,54,55,56,57,58,59]. However, the majority of these studies were conducted in model cellular systems (mainly in fibroblasts). In 1995, in searching for the “ovarian growth factor” in human ascites from ovarian cancer patients, Xu et al. published three seminal papers linking pathologic LPA to cancer (breast and ovarian cancer cells, as well as leukemia cells) [8,9,10]. Since then, the research on the relevance of LPA in cancer and human health is booming (Figure 1).

#### 2.1.2. Post Identification of LPA Receptors

Although G-protein-mediated LPA signaling pathways were identified as described above, molecular identification and cloning of LPA receptors in 1996 and the following years established the corner stones for rapid growth of LPA-related studies and targeting, as G-protein coupled receptors (GPCRs) represent targets for ~40% of pharmacological drug antagonists [123].

The first LPA receptor was identified and cloned in 1996 [60], which was followed by cloning and identification of a total of six LPA receptors, namely LPAR_1_/EDG2, LPAR_2_/EDG4, LPAR_3_/EDG7, LPAR_4_/purinergic G protein-coupled receptor P2Y9 (P2Y9/GPR23, LPAR_5_/GPR92, and LPAR_6_/P2Y5 [61,62,63,64,65,66,67,68,69,70,71]. Several additional G-protein coupled receptors (GPCRs) were also shown to be putative LPA receptors, including GPR87 [124,125], GPR35 [126], and P2Y10 [127]. However, they are less studied and/or not confirmed as LPA receptors. Moreover, the nuclear receptor peroxisome proliferator-activated receptor gamma (PPARγ) was identified as an intracellular LPA receptor [128,129,130,131]. PPARγ belongs to the nuclear receptor superfamily of PPARs (PPARα, PPARβ/δ, and PPARγ). PPARs play a role in inflammation and a variety of cancers which include prostate, breast, glioblastoma, neuroblastoma, pancreatic, hepatic, leukemia, and bladder and thyroid cancers [132], and they are mainly studied by using their natural and synthetic agonists or antagonists, including thiazolidinediones, different unsaturated fatty acids, and GW9662. The results are contradictory, with both pro- and anti-tumor roles of PPARγ reported [132]. LPA was identified as a new ligand for PPARγ in 2003 [60]. Until recently, LPA–PPAR*γ* studies were mainly limited to the vascular and metabolic processes [130]. We recently showed that LPA upregulates an oncogene *ZIP4* in epithelial ovarian cancer (EOC) cells, mainly via PPAR*γ*, and LPA’s cancer stem cell (CSC)-promoting activities are mediated by PPAR*γ* [133].

Another important milestone in LPA research was the identification of the major LPA-producing enzyme, autotaxin (ATX). Although the enzymatic activity of the lysophospholipase D in the production of LPA was described earlier in rat plasma [134], the gene encoding this enzyme for this activity was not known until 2002 [88,89].

The crystallization and structure determination for LPA GPCRs belonging to each of the two subclasses (EDG and purinergic receptors), as well as ATX [90,91,92,93,94], in recent years were pivotally important in design and development of anti-cancer reagents targeting them. In fact, Food and Drug Administration (FDA)-approved inhibitors against ATX and LPA monoclonal antibody entered into clinical trials for fibrosis [95] (Figure 1).

The functions/cellular effects of LPA (both physiological and pathological) are very broad, which were extensively reviewed [3,35,36,135,136,137,138]. The signaling pathways, mainly those mediated by LPA GPCR receptors, were also extensively studied and reviewed [3,35,61,135,136,138,139]. This review focuses on the challenges in moving bench LPA studies to clinical practice (bedside).

### 2.2. Challenges and Obstacles of LPA Clinical Applications in Cancer

#### 2.2.1. The Issues with LPA as a Marker for Cancer

We initially reported LPA as a potential marker for ovarian cancer (OC) [72], which is supported by blinded [22] and numerous independent studies [73,74,75,76,77,78,79,80,81]. LPA was also shown to be a biomarker for other gynecological cancers [72], as well as for gastric cancer [82].

However, we are facing several challenges in moving LPA as a cancer marker to clinical application [7]. These issues are tightly related to the biochemical nature of LPAs, which are metabolites, having a quick turnover time due to their producing and degradation enzymes, as well as several other potential factors [7,81,140,141]. Many epidemiological factors, such as diet, smoking, and drinking may also have significant effects on LPA levels detected, which are not always included in various studies. Technical issues are another major concern. These issues include many different lipid extraction, storage, and detection methods used, which may generate LPA artefacts [7,141].

The analytic methods for LPA were greatly advanced from earlier (1960–1990s) thin-layer and high-performance liquid chromatography-based analyses [8,142,143] to modern electrospray ionization tandem mass spectrometric (ESI-MS/MS) methods [73,83,84,85,86,87]. Another major technological advancement in LPA detection and targeting is the development of antibodies against LPA. LPA is not immunogenic, since all animal species produce LPA and LPA is very small (molecular weights ranging from 400 to 500 Da), lacking structural specificity to elicit a specific immune response. Nevertheless, Lpath Inc. successfully developed monoclonal LPA antibodies via their proprietary technique, which were used in research [144,145] and commercially available LPA enzyme-linked immunosorbent assay (ELISA) detecting kits (e.g.; Echelon Biosciences, T-2800s). These methods, however, have a limitation where it is not possible to distinguish individual LPA species as the ESI-MS/MS analysis does.

To overcome these obstacles, one possibility is to measure the levels of ATX, the key enzymes producing LPA [88,89], such as in the case of breast cancer and follicular lymphoma [146,147]. However, LPA levels are controlled by a complex array of enzymes and other conditions (see Section 2.2.2); therefore, ATX levels may not always correlate well to LPA levels. For example, while LPA levels are elevated in EOC [22,72,73,74,75,76,77,78,79,80,81], ATX levels are indifferent in control and EOC subjects [77,148].

It may be critical to develop more direct detection methods for LPA from human samples, such as a drop of fingertip blood on a filter paper, to avoid effects derived from variations in lipid extraction and storage conditions. Direct imaging/reporter-based methods may represent another direction to bypass the sample handling related artefacts.

#### 2.2.2. Targeting LPA Metabolism 

As mentioned above, LPA represents a group of compounds varying in their chemical linkage to the glycerol backbone, number of carbons, and number and location of double bonds, with their molecular weights between 400 and 500 Da [61,149]. In addition, several chemically closely related compounds, including sphigosine-1 phosphate (S1P) [7,21,150,151], cyclic phosphatidic acid (cPA) [152,153,154,155], and platelet-activating factor (PAF) [156,157,158], as well as other lysophospholipids [7,21] share similar, distinct, or opposing signaling and functions to LPA. While this review focuses on LPA, it is important to note that these additional lipids and their strong intertwining metabolic/catabolic pathways and interactions in function make targeting LPA much more complex and challenging [71].

LPA production and catabolism are controlled by a complex network of enzymes. Extracellular LPA is mainly produced by ATX and soluble phospholipase A_2_ enzymes (sPLA_2_s) [159]. Other PLA_2_s [160,161,162,163,164,165,166] and lipid phosphate phosphatase enzymes (LPPs) [5,167,168] play important roles in LPA generation and degradation, respectively. PLA_2_s are not only critical in generating the substrates for ATX to produce LPA, but they also generate LPA directly by acting on phosphatidic acid as its substrate [159,160,161,162,163,164,165]. To-date, among the 22 identified human PLA_2_s, at least 10 were studied in cancer, with most of them being aberrantly expressed in cancer [160] (Figure 2).

LPPs are major LPA catabolic enzymes. By removing the phosphate from LPA, they inactivate most of LPA’s biological effects [5,167,168]. Other LPA-related enzymes include several mono- or diacylglycerol kinases (MAGs and DAGs) involved in generating intracellular pools of LPA [126,169,170], and lysophospholipase transacylase (LLPT) or LPA acyltransferases (LPATs) inactivating LPA by converting it to phosphatidic acid [171,172,173] (Figure 2).

While depletion of ATX is embryonically lethal, postnatal decreases in the expression of ATX or LPPs produce little obvious phenotypic change, suggesting less toxicity is expected when targeting these enzymes [95]. Inhibitors against ATX and LPA monoclonal antibody entered into clinical trials for fibrosis, but are yet to do so for cancer [95]. Targeting LPPs was not tested clinically, although in vitro and preclinical studies support their anti-cancer roles [95,174].

At any rate, the complex array of enzymes and their regulations in LPA metabolism is a major obstacle in targeting LPA production. In addition, the enzymes shown in Figure 2 are also involved in the metabolism of other lipid molecules, further complicating the overall outcome. For example, ATX also generates cyclic phosphatidic acids (cPAs: naturally occurring analogs of LPA), which have anti-proliferative and anti-tumor activities [175,176] (Figure 2).

This situation is further complicated with the involvement of the tumor microenvironment (TME), which was recently reviewed extensively [7,95,135]. It was shown that ~40% of ATX in the body is produced by adipocytes, and this is increased further by inflammation in obesity linked to insulin resistance [95,177]. Cross-regulation of the immune/inflammation system, and the preferred adipose tissues for LPA production are emerging as critical targets for breast and multiple aggressive abdominal cancers, including colon, ovarian, and pancreatic cancers [7,95].

#### 2.2.3. Targeting LPA Receptors

GPCRs are the largest superfamily of receptors, with the identification of 865 human GPCR genes [178]. Compared to other plasma membrane receptor types with more specific ligand types, including receptor tyrosine kinases (RTKs), integrins, and ion transporters, ligands of GPCR cover very diverse chemicals, including amino acids, amine derivatives, peptides, proteins, lipid molecules, mechanical stimuli, and even ions, such as Ca^2+^, protons, and photons [179]. GPCRs are involved in almost all of the important physiological and many critical pathological processes. About 40% of drugs on the market act on GPCRs as agonists or antagonists [123]. The majority of LPA’s tumor-promoting activities are mediated by LPA GPCR receptors, naturally making them one of the most important targets.

One of the challenges in targeting LPA GPCR receptors is their complex array of G-protein coupling, resulting in multi-faceted outcomes. While most of the other individual GPCRs, including most of the best-studied β-adrenergic receptors, neurotransmitter receptors, and sensor GPCRs (olfactory, taste, and photosensory receptors) [180,181,182,183,184] couple to one or two specific types of trimeric G-protein, each LPAR couples to multiple G-proteins [61]. Further studies after the review in 2014 [61] showed single G-protein coupled LPAR_6_ and double G-protein coupled LPAR_3_ to couple to both G_12/13_ and G_i_ for LPAR_6_ [185,186,187] and G_13_, in addition to G_q_ and G_i_ for LPAR_3_ [188,189] (Figure 3).

Of interest, G_s_ coupling is involved in many essential physiological functions, ranging from cardiovascular effects mediated by adrenergic receptors to neurotransmission by dopamine and serotonin receptors, various hormonal effects by hormone receptors, energy and inflammation regulation mediated by purinergic receptors, and skin pigmentation regulation by melanocortin receptors [184,190,191,192,193,194,195,196,197]. In particular, all olfactory GPCRs, which consist of ~40% of all GPCRs in humans, are coupled to G_s_ [180,198]. However, G_s_ in general is involved in anti-cancer activities. 

While tumor-promoting activities are more consistently associated with LPAR_1–3_, which are all coupled to the G_i_/Ras/MAPK pathway [61], LPAR_4–6_ predominantly show anti-tumor activities. For example, in colon cancer cells, LPAR_1_ and LPAR_6_ positively and negatively regulate colony formation, respectively [199]. LPAR_4_ reduces cell proliferation, motility, and invasiveness in head and neck squamous cells [200]. In pancreatic cancer cells, downregulation of LPAR_4_ and LPAR_5_ enhanced the cell motility and colony formation activities [201]. LPAR_5_ inhibited the cell motility activity of sarcoma and endothelial cells [202]. These inhibitory effects are most likely associated with the predominate ability of LPAR_4–6_-mediated cAMP elevation via G_s_-coupling (such as in the case of LPAR_4_ and LPAR_6_) or a potentially G_s_-independent pathway to increase cAMP via LPAR_5_ [149]. Contradictory effects of LPAR_5_ were also shown to enhance cell proliferation and motility in rat lung and liver cancer cells [203], which may be related to its ability to couple to G_q_ and/or G_12/13_ [61]. On the other hand, the inhibitory effects of LPAR_5_ in cytotoxic T cells may actually have a pro-tumorigenic effect [204] (Figure 3).

Many LPA GPCR receptor agonists and antagonists were developed [61]. However, most, if not all, of them have cross-activities on more than one LPA receptor or other target [61], potentially complicating the outcomes using these inhibitors. Different LPARs are differentially expressed in different cancers and different individuals. In addition, the existence of non-GPCR LPA receptors, such as PPARγ, also needs to be considered. Studies using inhibitors against LPAR_1_, LPAR_1/3_, ATX, and LPA monoclonal antibodies recently entered clinical trials for fibrosis [95]. Cancer treatment using these reagents may be expected in the near future. However, more specific targeting of the particular tumor promoting LPAR(s) on an individual cancer and person-based manner is likely to be critical to make this targeting clinically beneficial.

As mentioned above, there are many different species of LPAs, which have preferences to bind to different LPA receptors. For example, LPAR_3_ preferentially binds to LPA with unsaturated fatty acids [51,205]. In addition, LPA GPCR receptors were shown to have ligands in addition to LPAs. For example, peptone (protein hydrolysates) and farnesyl pyrophosphate are agonists for LPAR_5_. GPR35 is a receptor for a number of naturally occurring lipids, including kynurenic 2-arachidonoyl LPA and lysophosphatidylinositol [126,149]. These issues are under-investigated, but may play significant roles in clinical practice.

#### 2.2.4. Targeting LPA Cross-Talk

##### 2.2.4.1. Cross-Talk between LPA Signaling and Other Cell Signaling Receptors

LPA elicits multiple and complex signaling pathways, which were extensively reviewed in recent years [21,31,48,61,95,206,207,208,209]. LPA signaling pathways intertwine with almost all other major cell signaling pathways. We postulate that this network, instead of an individual LPA signaling pathway, represents a more effective target. Hence, this review focuses on LPA cross-talk with other signaling molecules. These molecules are often more “targetable” with FDA-approved inhibitors in clinical trials.

The cross-talk between LPA and other signaling molecules was extensively demonstrated, covering virtually every type of cell plasma membrane receptors, including ligand-gated ion channels, receptor tyrosine kinases (RTKs), receptors with other enzymatic activities (serine or serine/threonine kinases and guanylyl cyclase enzymatic activities), other GPCRs, integrins, cytokine receptors, and T- and B-cell receptors, as well as intracellular receptors, such as PPARγ. Listed below are examples from these categories (Figure 4).

LPA stimulates and regulates several ion channels, including the Ca^2+^ and Ca^2+^-activated potassium channels, and the Na^+^/H^+^ exchanger 3 (NHE3) via the LPAR_5_ receptor, which also involves the epidermal growth factor receptor (EGFR) [210,211]. LPA also regulates glucose transporters in skeletal muscle and adipose tissue [212]. We recently showed that LPA upregulates ZIP4 (a zinc transporter) expression mainly via PPAR*γ* [133] (Figure 4).

The cross-talk between receptor tyrosine kinase (RTK)–GPCR signal complexes is a focal point for the study of integration of cell signaling, which plays an important role in signal transduction [213]. The cross-talk between LPA and EGFR is the best studied [214,215,216,217,218,219]. LPA also regulates and/or transactivates platelet derived growth factor receptor (PDGFR) [220,221,222,223,224], tropomyosin receptor kinase A (TrkA), the receptors for nerve growth factor (NGF) [225], Toll-like receptors [226], and c-Met, the receptor for hepatocyte growth factor [227].

LPA inhibits the natriuretic peptide-induced generation of cGMP via a non-receptor tyrosine kinase Csk [228,229,230]. The best example of LPA’s effect on non-membrane receptors is its functions with regard to Src family kinases [231,232,233,234]. In addition, LPA regulates cytokines, such as IL-6, and its downstream signal transducers and activators of transcription (Stat) signaling molecules [235].

LPA interacts with other GPCR receptors. Free fatty-acid receptors (FFARs; FFA1 and FFA4) have a potential negative cross-talk between LPA receptors and EGF receptors [217,236]. LPA stimulates endothelin (a GPCR ligand) expression and production in vascular smooth muscle cells [237]. In addition, a cross-talk between the LPAR–G_13_/p115RhoGEF/RhoA pathway and the β2-adrenergic receptor/G_s_/adenylyl cyclase pathway was reported [238]. LPA also cross-talks with α1 adrenoceptors [239]. At physiological concentrations, LPA is capable of modulating opioid receptor binding [240].

There are close interactions between two oncogenic lysolipids, LPA and S1P, in their overlapping signaling pathways and/or directly in their receptors [241]. These two lipids can also cross-talk via ATX [242,243]. Transforming growth factor beta (TGFβ) may play a role in the LPA–S1P cross-talk [244]. LPA upregulated expression of the cyclin-dependent kinase inhibitor p21(Waf1) in a TGFβ-dependent manner [245]. Cross-talk between TNF-α and LPA results in the amplification of COX-2 protein expression via a conserved protein kinase D (PKD)-dependent signaling pathway [246]. Hisano et al. used a genome-wide CRISPR/dCas9-based GPCR signaling screen to identify that LPAR_1_ is an inducer of S1PR_1_/β-arrestin coupling. This interaction promotes the porous junctional architecture of sinus-lining lymphatic endothelial cells and enables efficient lymphocyte trafficking [247]. The functional link between LPA and integrins was established. Active integrin β1 is required for migration of fibroblastic cells [248]. Laminin, but not other extracellular matrix proteins, induces LPA production in ovarian cancer cells via a β-integrin [164]. LPA induces αvβ6-integrin-mediated TGFβ activation via the LPAR_2_ and the small G_q_ [249]. LPA upregulates integrins [250,251], and integrin signaling regulates the nuclear localization and function of the LPAR_1_ in mammalian cells [252]. Moreover, LPA-induced RhoA activation integrates the functions of integrins [251,253] and integrin α6β4 promotes expression of ATX in breast cancer cells [254]. Most noticeably, ATX directly binds to several integrins [91,255], producing LPA close to the cell membrane [256] (Figure 4).

LPAR_5_ functions as an inhibitory receptor able to negatively regulate T-cell receptor (TCR) signaling [204]. LPAR_5_ also inhibits B-cell receptor (BCR) signal transduction via a G_α12/13_/Arhgef1 pathway [257]. On the other hand, LPA augments IL-13 secretion from T cells via induction of submaximal T-cell activation [258].

The cross-talk can be mono- or bidirectional and can be either positive or negative cross-talk, dependent on the type of interaction, the cell types, and the biological effects involved [259]. For example, while LPA transactivates nerve growth factor signaling via the TrkA receptor, the latter also uses a G-protein-mediated mechanism to regulate the p42/p44 MAPK pathway [260]. The bidirectional regulation between LPA and integrins is mentioned above (Figure 4).

It is important to note that LPA is involved in several stem cell/cancer stem cell (CSC) signaling pathways. The ATX–LPA signaling pathway is recognized as a critical new player in CSC [48]. LPA is involved in classical stemness pathways, such as the Wnt, Notch, and Hippo pathways [189,261,262,263,264,265].

##### 2.2.4.2. The Molecular Mechanisms of LPA Cross-Talks

LPA cross-talks with other signaling molecules at many different levels with divergent mechanisms. Firstly, interactions are through direct binding/interactions. Homo- and heterodimerization of LPA/S1P receptors, ovarian cancer G protein coupled receptor-1 (OGR1) and GPR4, was shown using LacZ complementation assays [266]. LPA receptors form homo- and heterodimers within the LPA receptor subgroup and heterodimers with other receptors, such as S1PR_1–3_ and GPR4. Interestingly, it was shown that LPA remarkably enhances, through the LPAR_1_/G_i_ protein, the OGR1-mediated vascular actions to acidic pH [267]. These results suggest that targeting dimerization may be an effective way to block the signaling mediated by the receptors. Although GPCR dimerization was known for many years, this is an under-investigated area and warrants further investigation [266] (Figure 4).

Secondly, transactivation is mediated via enzymatic activities regulating phosphorylation and/or ligand processing. LPA induces EGF receptor transactivation through metalloproteinase (MMP) and a disintegrin and metalloproteinase (ADAM)-catalyzed membrane shedding of heparin-binding EGF and autocrine/paracrine activation of EGF [231,268,269], and EGF can also modulate LPAR_1_ function and the phosphorylation state [268] (Figure 4).

Thirdly, an LPA-regulated transcriptome is involved. LPA regulates many cytokines, including IL-6, IL-8, growth-regulated oncogene (GRO)-α [19,20,270,271], and cytokine leukemia inhibitory factor (LIF) [241]. IL-6 mediates the LPA cross-talk between stromal and epithelial prostate cancer cells [272]. LPA-induced macrophage migration inhibitory factor (MIF) promotes both tumor cell growth and angiogenesis via both the Ras/MAPK and Ras–Akt/PI3K signaling pathways [273]. IL-6 exerts its biological activities through two molecules: IL-6R (IL-6 receptor) and gp130 [274]. Moreover, gp130-mediated Janus kinase (JAK)/signal transducer and activation of transcription 3 (STAT3) is required for ATX expression in adipocytes [177] (Figure 4). LPA stimulates the expression of CSC-associated genes, including *OCT4*, *SOX2*, *SOX9*, *ALDH1*, and drug transporters [133,275,276], with most of these gene products being functionally involved in CSC.

Fourthly, downstream signaling pathway interactions play important roles. The signaling pathways involved in LPA cross-talk include, but are not limited to the PI3K/Ras [277], the mitogen-activated protein kinase (MAP kinase) [277,278], the focal adhesion [119], the Wnt, integrin, the Rho/Rock, and the YAP pathways [279,280], reactive oxygen species (ROS), the DNA repair pathway, and the glycolytic pathway [27], as well as the Rho–cAMP interaction [281] (Figure 4).

Finally, other signaling molecules may regulate metabolic enzymes for LPA and other lipid molecules. Neurotransmitters, cytokines, and growth factors regulate the activity of a key set of lipid-metabolizing enzymes, such as phospholipases, to affect LPA and other lipid signaling molecules [282]. In addition, an acylglycerol kinase that produces LPA modulates cross-talk with EGFR in prostate cancer cells [283].

The targeting of one or more of these cross-talks and/or the major LPA downstream signaling pathways may be critical and/or more efficient than targeting LPA or LPAR directly. For example, the FDA recently approved the first PI3K inhibitor for breast cancer treatment. The challenges are identifying one or more driver targets at the level of individual cancer type and individual patient.

#### 2.2.5. Targeting Tumor–Stromal Interactions in the TME

Targeting the tumor-prone microenvironment gained increasing attention in recent years [7]. Although ATX can be produced directly by cancer cells, such as in melanomas, glioblastomas, and thyroid tumors [95], it may be mainly produced by stroma cells, as ~40% of ATX in the body is produced by adipocytes, and this is increased further by inflammation in obesity linked to insulin resistance [95]. In addition, macrophage-derived phospholipase A_2_ (PLA_2_), which is a soluble PLA_2_, produces extracellular LPA and is involved in EOC and associated with early relapse of EOC [284,285] (Figure 5).

While S1P’s functions in the immune system were extensively studied, and the S1P receptor axis represents an obligatory signal for trafficking of immune cells [34], the role of LPA in the immune system is less studied [34]. LPA affects TCR and BCR as mentioned in Section 2.2.4.1, and LPA converts monocytes into macrophages in both mice and humans [286] (Figure 5). In addition, ATX represents a connecting point for both S1P and LPA, since it is an enzyme producing both S1P and LPA [287]. More interestingly, a recent study showed that S1P/S1PR4 and ATX/LPA/LPAR_5,6_ appear as critical axes for immune infiltrates [34]. These were robust differences in sphingolipid/LPA-related checkpoints and the drug response. Genes including *CD68* (a monocyte/macrophage marker), *LPAR_3_* (a LPA receptor), *SMPD1* (sphingomyelin phosphodiesterase 1 that converts sphingomyelin to ceramide), *PPAP2B* (LPP3, a phosphatidic acid phosphatase, converting phosphatidic acid to diacylglycerol and LPA to monoacylglecerol [288]), and *SMPD2* (sphingomyelin phosphodiesterase 2, with lysophospholipase activity) emerged as the most prognostically important markers. In particular, alignment of data across a variety of malignancies (over 600 different neoplasm categories) revealed specific preference for ovarian carcinoma [34]. It is interesting that ATX, LPAR_1_, and LPAR_5_ are higher in the immune-high tumor (Cd14-, Cd68-, Cd164-, and Cd3E-high) group, but LPAR_2.3_ are higher in the immune-low group [34], suggesting the complex regulatory roles of the ATX–LPA axis in the tumor–immune system interaction.

## 3. Conclusions

While LPA and/or the ATX–LPA axis are generally accepted as important targets for cancer, we are still facing several major obstacles to move targeting to clinical practice as presented above. Personalized medicine is now well accepted conceptually to treat highly heterogenic diseases, such as cancers. The complex LPA metabolism, its receptor, and its signaling systems suggest that detecting signatures/networks, rather than individual gene/protein/lipid expression, from individual patients will likely become necessary to develop effective treatments for highly heterogenic diseases, such as cancer. Bioinformatics/systems biological analyses become very important to achieve such goals, which provide unprecedented scale and depth of knowledge and perspectives in cancer research, as well as directions for translational and clinical applications. These studies use multi-modular integrative approaches consolidating large amount of data from gene expression profiling, next-generation sequencing, -omics studies, prognostic/predictive modeling, and functional studies for cancer [295,296]. The most prominent example is the Pan-cancer Atlas analyses organized by the National Institute of Health (NIH), which is based on The Cancer Genome Atlas (TCGA) data of over 11,000 tumors from 33 of the most prevalent forms of cancer, using comprehensive, in-depth, and interconnected bioinformatics analyses (https://www.cell.com/pb-assets/consortium/ pancanceratlas/pancani3/index.html?code=cell-site). One such study related to LPA identified novel sphingolipid–LPA immune checkpoints and networks underlying tumor immune heterogeneity and disease outcomes, as mentioned in the Section 2.2.5 [34]. This type of study holds great promise for delivering novel stratifying and targeting strategies [34].

It is important to note that, although targeting LPA signaling appears to be an attractive approach for cancer treatment, it is unlikely that this will be effective as monotherapy. One of the major challenges in LPA targeting will be to discover the magic modalities to be effective in cancer treatment, which are likely to be different for different cancers and even for different individuals. Synthetic lethality studies led to the clinical benefits of using poly (ADP-ribose) polymerase (PARP) inhibitors in OC and other cancers, which is a very significant step forward in the field [297]. Similar studies are urgently needed for LPA targeting through logical and empirical screening. As an interesting note, accumulated data suggest that LPA is a very strong inducer for cell migration, invasion, and tumor metastasis, but a rather weak proliferation stimulus [4,23,25]. On the other hand, growth factors, such as EGF and insulin growth receptor (IGF), possess strong cell proliferation, but rather weak chemotactic activities. Proper combinational inhibition of these two sets of critical activities in cancer cells simultaneously may represent an effective strategy (Figure 4). Both in vitro and in vivo studies in these areas are highly needed to move the field forward.

Taken together, although we are still facing a number of obstacles, the large body of data generated from the last 3–4 decades from thousands of papers provided solid evidence and rationale to target the LPA network. Rapid development of biotechnological tools for analyses of LPA, as well as its functions and mechanisms, and for processing the vast amount of data provided and will continue to furnish essentials for the field to leap ahead and eventually improve patient outcomes.

## Figures and Tables

**Figure 1 cancers-11-01523-f001:**
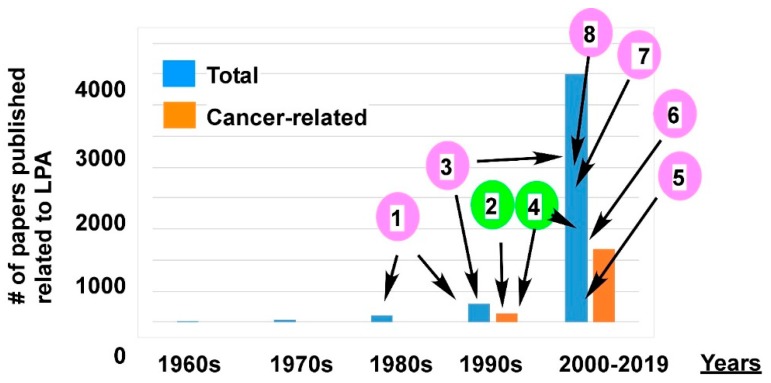
Lysophosphatidic acid (LPA)-related papers published in decades and the milestones in LPA research. The blue bars are the total number of LPA-related papers published in each decade. The orange bars are the numbers of LPA studies related to cancer. The pink circled numbers are milestones related to LPA research in general, and the green circled numbers are cancer-related milestones. (1) LPA’s mitogen and growth factor like activity, as well as G-protein-mediated signaling mechanisms were discovered in the late 1980s and early 1990s [35,53,54,55,56,57,58,59]. (2) In 1995, the pathological significance of LPA in cancer was first reported [8,9,10]. (3) From 1996 to 2009, six LPA G-protein coupled receptors (GPCRs) were identified and cloned [60,61,62,63,64,65,66,67,68,69,70,71]. (4) From 1998 to the present, LPA as a putative cancer marker was reported [22,72,73,74,75,76,77,78,79,80,81,82]. (5) From 2000 to the present, new technologies, including the electrospray ionization tandem mass spectrometry (ESI-MS/MS) methods, were developed for LPA analyses [73,83,84,85,86,87]. In addition, LPA antibodies were developed and further improved. In 2008, Lpath Inc. successfully humanized an anti-LPA antibody. (6) In 2002, the major LPA-producing enzyme ATX was identified and cloned [88,89]. (7) From 2011 to 2017, ATX and LPA G-protein coupled receptors (GPCRs) were crystalized with their structures determined [90,91,92,93,94]. (8) From 2013 to the present, FDA-approved ATX and LPA receptor (LPAR) inhibitors entered clinical trials for fibrosis [95].

**Figure 2 cancers-11-01523-f002:**
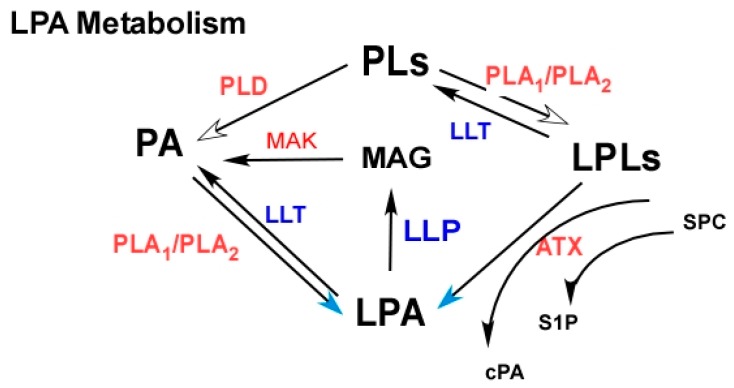
LPA metabolism as potential targets. Phospholipids (PLs), phosphatidic acid (PA), lysophospholipids (LPLs). The enzymes in red color, autotaxin (ATX), phospholipase A_1_ (PLA_2_), phospholipase D (PLD), and monoacylglycerol kinase (MAK), need to be inhibited to reduced LPA. ATX inhibitors are currently in clinical trials. The enzymes in blue, lipid phosphate phosphatase enzymes (LPPs), lysophospholipase transacylase (LLPT), or LPA acyltransferase (LPAT), need to be enhanced to increased LPA degradation. However, these enzymes are also involved in the metabolism of other lipid molecules, and the overall outcome may be complex. ATX may have multiple functions. It also produces sphingosine-1 phosphate (S1P) from sphingosylphosphorylcholine (SPC) and cyclic phosphatidic acids from lysophospholipids (LPLs). Cyclic PAs (cPAs) have anti-tumor activities [176].

**Figure 3 cancers-11-01523-f003:**
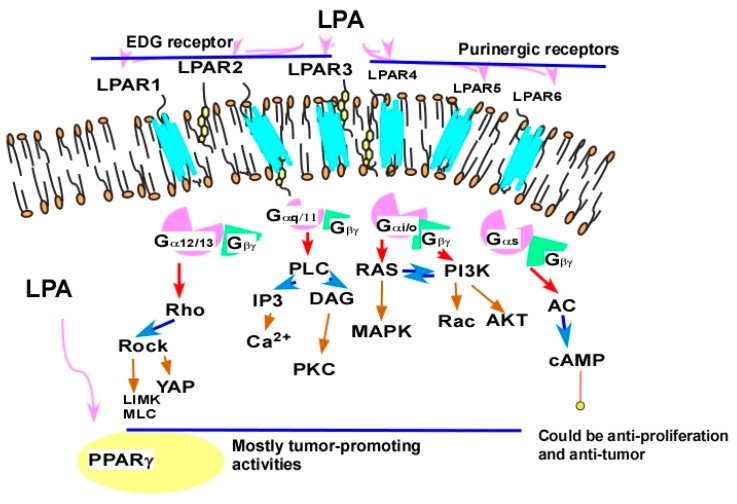
LPA receptors as targets. In general, the EDG family LPA receptors (LPAR_1–3_) are coupled to G_i_, G_q_, and G_12/13_ proteins [61,189] and are more involved in tumor-promoting activities. The purinergic family LPA receptors (LPAR_4–6_) are all coupled to G_12/13_ and other trimeric proteins [61,185,186,187]. Their anti-tumor effects may be mediated by their ability to elevate cyclic adenosine monophosphate (cAMP) levels [149].

**Figure 4 cancers-11-01523-f004:**
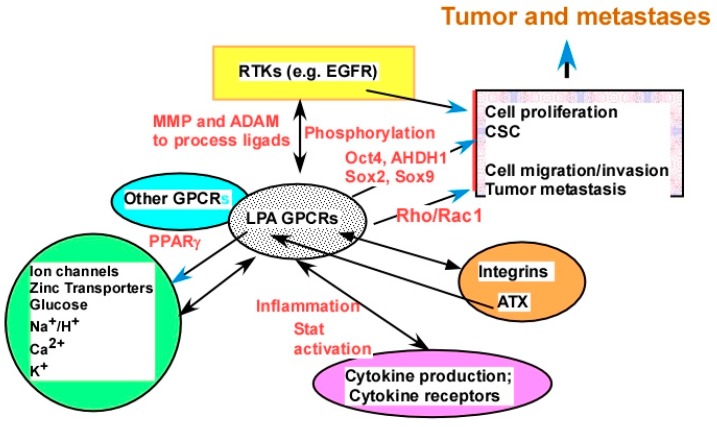
LPA cross-talk as potential targets. LPA interacts with major types of plasma membrane receptors, including ion channels, metal ion transporters, other transporters, receptor tyrosine kinases (RTKs), other GPCRs, integrins, and cytokine receptors. Examples from each category of receptors are discussed in the Section 2.2.4. Certain potential mechanisms of cross-talk are presented by words in red, including ligand production and/or processing, receptor phosphorylation, and production of downstream molecules mediating the cross-talk.

**Figure 5 cancers-11-01523-f005:**
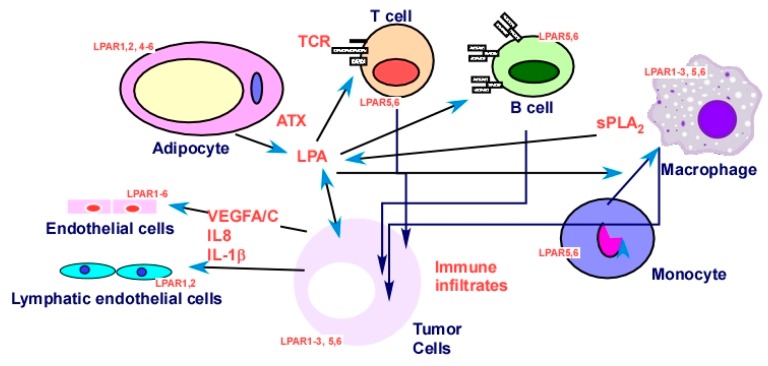
LPA in tumor cells and in the tumor microenvironment (TME). Tumor, stromal, and immune cells in the TME express LPA receptors, and they produce and/or respond to LPA [34,119,185,202,247,256,289,290,291,292,293,294]. The overall effects produce a tumor-promoting environment as detailed in Section 2.2.5 and in recent reviews [7,21,135].

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
