# Peer review of "Targeting Lysophosphatidic Acid in Cancer: The Issues in Moving from Bench to Bedside"

_cancers, 2019, doi:10.3390/cancers11101523_

Round 1

Reviewer 1 Report

This review article came from Dr. Yan Xu, an expert in, who does LPA research for many years. The manuscript is good in general but could be improved.

Abstract could be written to reflect the key points of the manuscript, focusing on scientific values and insights into the therapeutic implications in cancers by targeting LPA.

In addition to summarizing what has been reported or studied, the author could provide a deep analysis of current findings and insights into potential clinical applications.

The therapeutic potential of LPA in tumor microenvironment should also be discussed and emphasized including effect of LPA in immunity and angiogenesis.

Citations are better updated, for example, reference 1, Weinberg should have an updated review in Cell 2011 and the citations should be double checked and updated. A seminal paper from Mill’s group was published in Cancer Cell and other studies about LPA breast cancer progression (doi: 10.1016/j.ccr.2009.03.027; 10.18632/oncotarget.15123;), which described LPA in breast cancer progression and other cancer types (doi: 10.3390/cancers11070958; 10.1158/1078-0432.CCR-13-1284; 10.1016/j.cellsig). Many studies demonstrate the impact of LPA in angiogenesis, which is one of hallmarks of cancer and potential therapeutic target (doi: 10.1002/cncr.24907; 10.1161/ATVBAHA.116.307421; 10.1182/blood-2010-12-326017).

Line 180-181, the author need provide a citation (s) for the statement. Also please double check other statements.

Is it possible to have a smooth transition from the text body to conclusion?

English needs to be edited and improved. For example, in line 87 “LPA includes hypertension and other cardiovascular effects”. It is not clear what the author meant.

Author Response

This review article came from Dr. Yan Xu, an expert in, who does LPA research for many years. The manuscript is good in general but could be improved.

Abstract could be written to reflect the key points of the manuscript, focusing on scientific values and insights into the therapeutic implications in cancers by targeting LPA. In addition to summarizing what has been reported or studied, the author could provide a deep analysis of current findings and insights into potential clinical applications.

The Abstract is revised to reflect the key points of the manuscript as recommended.

The therapeutic potential of LPA in tumor microenvironment should also be discussed and emphasized including effect of LPA in immunity and angiogenesis. Citations are better updated, for example, reference 1, Weinberg should have an updated review in Cell 2011 and the citations should be double checked and updated. A seminal paper from Mill’s group was published in Cancer Cell and other studies about LPA breast cancer progression (doi: 10.1016/j.ccr.2009.03.027; 10.18632/oncotarget.15123;), which described LPA in breast cancer progression and other cancer types (doi: 10.3390/cancers11070958; 10.1158/1078-0432.CCR-13-1284; 10.1016/j.cellsig). Many studies demonstrate the impact of LPA in angiogenesis, which is one of hallmarks of cancer and potential therapeutic target (doi: 10.1002/cncr.24907; 10.1161/ATVBAHA.116.307421; 10.1182/blood-2010-12-326017).

The 2011 Cell paper has been cited in the original manuscript (#2). Other citations have been carefully rechecked and many of them have been updated. This review focuses on the issues pertinent to translating LPA targeting to the clinics. It does not intend to cover every aspect of LPA’s actions, which have been intensively reviewed in recent years. For examples, the author has recently reviewed “Lysophospholipid Signaling in the Epithelial Ovarian Cancer Tumor Microenvironment”, including angiogenesis (PMID: 29987226). Yu Hisano and Timothy Hla have extensively reviewed “Bioactive lysolipids in cancer and angiogenesis” (PMID: 30048709). These publications were mentioned in the previous version or have been added in the revision.

Line 180-181, the author need provide a citation (s) for the statement. Also please double check other statements.

The citations have been carefully rechecked and many of them have been updated.

Is it possible to have a smooth transition from the text body to conclusion?

A transitional sentence has been added at the beginning of the Discussion section.

English needs to be edited and improved. For example, in line 87 “LPA includes hypertension and other cardiovascular effects”. It is not clear what the author

English has been carefully re-edited by a native English speaker: Mr. Kevin McClelland.

Reviewer 2 Report

Dear authors

The authors have submitted a well-organized and educational review on lysophosphatidic acid (LPA) in cancer. The introduction, bench works, and  clinical implications are well discussed. Personally I recommend the acceptance, although I have a few minor comments as follows: 

1.The authors focus on the issues with perspectives to improve the LPA translational progress in cancer research. However, tumor lymphangiogenesis and metastases are major concerns and prognostic factors in cancer patients. However, it's a pity that the authors didn't discuss or mention about tumor lymphangiogenesis. In LPA-related lymphangiogenesis researches, there are distinguished published articles recommended to be cited and discussed. Here are two following recent articles personally suggested.

Lin YC, Chen CC, Chen WM, Lu KY, Shen TL, Jou YC, Shen CH, Ohbayashi N, Kanaho Y, Huang YL, Lee H. LPA1/3 signaling mediates tumor lymphangiogenesis through promoting CRT expression in prostate cancer. BBA-MOL CELL BIOL L. 2018 Oct;1863(10):1305-1315.

Wu PY, Lin YC, Huang YL, Chen WM, Chen CC, Lee H. Mechanisms of lysophosphatidic acid-mediated lymphangiogenesis in prostate cancer. Cancers. 2018 Oct; 10(11):pii:E413.

2. The authors briefly mentioned the close interactions between two oncogenic lysolipids, LPA and S1P in their overlapping signaling pathways and/or directly in their receptors. Currently it's also a promising issue on lymphangiogenesis, for example Hisano Y et al (J Exp Med. 2019 Jul 1;216(7):1582-1598.) has recently found cross-talk between LPAR1 and S1PR1 promotes the porous junctional architecture of sinus-lining lymphatic endothelial cells, which enables efficient lymphocyte trafficking. The point between LPA and S1P cross interaction in lymphangiogensis might be also organized with my comment 1.

Author Response

The authors have submitted a well-organized and educational review on lysophosphatidic acid (LPA) in cancer. The introduction, bench works, and clinical implications are well discussed. Personally I recommend the acceptance, although I have a few minor comments as follows: 

1.The authors focus on the issues with perspectives to improve the LPA translational progress in cancer research. However, tumor lymphangiogenesis and metastases are major concerns and prognostic factors in cancer patients. However, it's a pity that the authors didn't discuss or mention about tumor lymphangiogenesis. In LPA-related lymphangiogenesis researches, there are distinguished published articles recommended to be cited and discussed. Here are two following recent articles personally suggested.

Lin YC, Chen CC, Chen WM, Lu KY, Shen TL, Jou YC, Shen CH, Ohbayashi N, Kanaho Y, Huang YL, Lee H. LPA1/3 signaling mediates tumor lymphangiogenesis through promoting CRT expression in prostate cancer. BBA-MOL CELL BIOL L. 2018 Oct;1863(10):1305-1315. 30053596

Wu PY, Lin YC, Huang YL, Chen WM, Chen CC, Lee H. Mechanisms of lysophosphatidic acid-mediated lymphangiogenesis in prostate cancer. Cancers. 2018 Oct; 10(11):pii:E413.

2.The authors briefly mentioned the close interactions between two oncogenic lysolipids, LPA and S1P in their overlapping signaling pathways and/or directly in their receptors. Currently it's also a promising issue on lymphangiogenesis, for example Hisano Y et al (J Exp Med. 2019 Jul 1;216(7):1582-1598.) has recently found cross-talk between LPAR1 and S1PR1 promotes the porous junctional architecture of sinus-lining lymphatic endothelial cells, which enables efficient lymphocyte trafficking. The point between LPA and S1P cross interaction in lymphangiogensis might be also organized with my comment 1.

As mentioned above, this review focuses on the issues pertinent to translating LPA targeting to the clinics. It does not intend to cover every aspect of LPA’s actions, which have been intensively reviewed in recent years. Nevertheless, we have added lymphangiogenesis in the first paragraph of the Introduction with the recommended additional references. The ref. related to the crosstalk has been added to the revised Section 2.2.4.

Reviewer 3 Report

The author has tried to cover a lot of ground in this review...probably too much. The title indicates that it is supposed to be about the clinical applications of LPA as a target in cancer, but in fact it reviews almost every aspect of LPA function. As a result, some of the sections are too superficial.

The discussion of how the field of LPA research has expanded over the years is interesting and makes the review unique.

Figure 2 has way too much going on...there are four major figures in one. Only the first is discussed in detail in the legend. I suggest that they be separated if all are retained.

While the number of reference citations is huge, the references are not always that recent. In some sections, the discussion seems a bit outdated as a result.

The sections concerning signal transduction and receptor cross-talk need updating if they are to be retained in the review.

The conclusion is thoughtfully written, and overall the author does take care to address important "issues" as stated in the title.

There is a major problem with English usage throughout the manuscript, making it difficult to follow in some sections.

Author Response

The author has tried to cover a lot of ground in this review...probably too much. The title indicates that it is supposed to be about the clinical applications of LPA as a target in cancer, but in fact it reviews almost every aspect of LPA function. As a result, some of the sections are too superficial.

In this revision, we emphasize that our focus is on the translational issues.  In addition, the crosstalk section has been updated.

The discussion of how the field of LPA research has expanded over the years is interesting and makes the review unique.

Figure 2 has way too much going on...there are four major figures in one. Only the first is discussed in detail in the legend. I suggest that they be separated if all are retained.

The original Fig. 2 has been split to four figures (Figs. 2-5) and legends to each figure have been revised.

While the number of reference citations is huge, the references are not always that recent. In some sections, the discussion seems a bit outdated as a result.

Many of early stage citations are intentionally used to reflect the history when the specific function/signaling was originally/first reported. At any rate, the references have been further updated in this revision.

The sections concerning signal transduction and receptor cross-talk need updating if they are to be retained in the review.

For the original “2.2.4. Targeting LPA signaling pathways and cross talks”, I have changed the title to be more focused on the cross-talks with updates. In addition, the original intention was to give a few examples for LPA’s cross-talk with each category of receptor and/or signaling molecules, but not to comprehensively cover all of the major studies and mechanisms in the field. Nevertheless, references are updated in the revision.

The conclusion is thoughtfully written, and overall the author does take care to address important "issues" as stated in the title.

There is a major problem with English usage throughout the manuscript, making it difficult to follow in some sections.

English has been carefully re-edited by a native English speaker: Mr. Kevin McClelland.

Round 2

Reviewer 3 Report

The authors appear to have addressed many of the comments of the reviewers. While the changes have improved the manuscript, extensive English editing is still needed. For example, it should always be "cross-talk" and not "cross-talks". The title of the review is awkwardly worded...better wording of the last phrawe might be, "...: Issues in moving from bench to bedside." The reference list is extensive so the literature coverage should be adequate. As noted previously, the signal transduction sections are relatively weak but are not really the main focus of the review.

Author Response

Changes have been made according to the reviewer's comments.